# Design of Geraniol-Loaded Nanocapsules for Use Against *Salmonella* Infantis: Evaluation in an In Vitro Poultry Model

**DOI:** 10.3390/pharmaceutics17070840

**Published:** 2025-06-27

**Authors:** Karla S. Garcia-Salazar, Hector J. Leon-Solano, Jesus A. Maguey-Gonzalez, Juan D. Latorre, Raquel López-Arellano, Elvia A. Morales Hipólito, Roberto Díaz-Torres, Miguel Morales Rodríguez, Alma Vázquez-Durán, Guillermo Tellez-Isaias, Abraham Méndez-Albores, Bruno Solis-Cruz, Daniel Hernandez-Patlan

**Affiliations:** 1Nanotechnology Engineering Division, Polytechnic University of the Valley of Mexico, Tultitlan 54910, Mexico; karla.garcia.salazar@upvm.edu.mx (K.S.G.-S.); hector.leon@upvm.edu.mx (H.J.L.-S.); 2Division of Agriculture, Department of Poultry Science, University of Arkansas, Fayetteville, AR 72701, USA; jm201@uark.edu (J.A.M.-G.); jl115@uark.edu (J.D.L.); 3Laboratory 5: Laboratorio de Ensayos de Desarrollo Farmacéutico, Multidisciplinary Research Unit, Superior Studies Faculty at Cuautitlan (FESC), National Autonomous University of Mexico (UNAM), Cuautitlan Izcalli 54714, Mexico; lopezar@unam.mx (R.L.-A.); eadriana_mh@comunidad.unam.mx (E.A.M.H.); 4Multidisciplinary Research Unit, Facultad de Estudios Superiores Cuautitlán, Universidad Nacional Autónoma de México, Carretera Cuautitlán-Teoloyucan Km. 2.5, San Sebastián Xhala, Cuautitlán Izcalli 54714, Mexico; diaztorres_r@cuautitlan.unam.mx; 5Departamento de Sistemas de Información y Comunicaciones, División de Ciencias Básicas e Ingeniería, Universidad Autónoma Metropolitana—Unidad Lerma, Lerma de Villada 52005, Mexico; m.morales@correo.ler.uam.mx; 6Unidad de Investigación Multidisciplinaria L14-A1 (Ciencia y Tecnología de Materiales), Facultad de Estudios Superiores Cuautitlán (FESC), Universidad Nacional Autónoma de México (UNAM), Cuautitlán Izcalli 54714, Mexico; almavazquez@comunidad.unam.mx (A.V.-D.); albores@unam.mx (A.M.-A.); 7Gut Healt LLC, Fayetteville, AR 72703, USA; gtellez@uark.edu

**Keywords:** geraniol, solvent-free polymeric nanocapsules, *Salmonella* Infantis, antimicrobial effect, interaction studies, FTIR-ATR, in vitro models, poultry industry

## Abstract

**Background/Objectives***: Salmonella* Infantis (*S*. Infantis) is a bacterium that has gained importance in public health over the last decade due to its high pathogenicity and resistance to antibiotics. Therefore, the objective of the present study was to present key considerations for the design and development of geraniol-loaded nanocapsules for its delivery in the drinking water or feed of broiler chickens and to evaluate its potential as an antimicrobial agent against *S*. Infantis using a standard in vitro microplate assay and a model that simulates the pH and feed conditions of the crop of broiler chickens. **Methods**: Using a 3^k^ factorial experimental design, geraniol nanocapsule-based formulations were selected, and their antimicrobial activity was evaluated in in vitro models. **Results**: The results demonstrated that geraniol alone exhibits antimicrobial action against *S*. Infantis mainly due to its lipophilicity, hydrophobicity and the presence of the hydroxyl group found in its chemical structure, but when formulated in nanocapsular systems, the interaction of its components tends to reduce its antimicrobial action, especially the mixture of Tween 80:Span 80 and Miglyol^®^ 810N. Furthermore, the use of the in vitro model that simulates the crop of broiler chickens demonstrated that the formulation also has interactions with the feed components, completely nullifying the antimicrobial action of geraniol compared to that obtained in the in vitro microplate model. **Conclusions**: Preformulation studies during the development of nanocapsule-based formulations should be considered for the correct selection of the components of a formulation to ensure its effectiveness, without only considering the physicochemical and stability properties of these as is frequently seen in studies.

## 1. Introduction

Non-typhoidal *Salmonella* infections are considered a major public health threat, primarily resulting from the consumption of poultry products contaminated with *Salmonella* enterica subspecies enterica serotypes Enteritidis (*S.* Enteritidis) and Typhimurium (*S.* Typhimurium) [1], with an estimated 94 million cases of infection and 215,000 deaths annually [2]. However, *Salmonella* enterica subspecies enterica serovar Infantis (*S*. Infantis) has now become an increasingly prevalent serovar worldwide [3]. Until 2019, *S*. Infantis was considered the sixth most frequent serotype of human salmonellosis in the USA, while in the EU it was the fourth most important, but since 2014, it has been the number 1 serotype isolated in broiler chickens and their derivatives, which makes it a foodborne pathogen of public health importance [4]. Although the main route of transmission of this pathogen to humans is through contaminated food of poultry origin, studies have also described its transmission through water, horizontal transmission between people, contact with infected animals and through the environment, so its prevalence has increased [5]. Furthermore, it has been reported that this bacterium is resistant to multiple drugs and has greater pathogenicity due to the presence of an emerging plasmid that encodes virulence factors and genes for resistance to antibiotics and even mercury [6].

In this context, and considering the regulatory frameworks of various global authorities regarding antibiotic use, there is an urgent need to research new viable alternatives to their use to address the problems of bacterial resistance. Thus, several alternatives have been studied for the prophylaxis, control, and treatment of bacterial infections in animals, including vaccination, the use of mono- and polyclonal antibodies, antimicrobial peptides, probiotics, prebiotics, bacteriophages, lysins, enzymes, and certain phytochemicals [7]. Of the above alternatives, phytochemicals offer a viable alternative to combat bacterial resistance problems since they have shown excellent antimicrobial and antioxidant properties [8,9]. Within this group are essential oils, bioactive natural compounds derived from plants, which have shown excellent antimicrobial, antioxidant, and anti-inflammatory properties [10]. Although numerous essential oils have been reported with primarily antimicrobial properties against Gram-positive and Gram-negative bacteria, in the present study, geraniol—a monoterpene alcohol found in essential oils of plants such as rose, ginger, lemon, orange, lavender and cannabis—was used because, after a systematic review, it was found to be the one with the best antimicrobial properties, low toxicity and high safety, which is why it is considered a compound generally recognized as safe (GRAS) by the Food and Drug Administration (FDA) [11,12]. However, limitations such as its poor aqueous solubility which results in poor bioavailability, its low stability since it hydrolyzes rapidly, as well as its intense flavor and odor, have conditioned its application in the poultry industry [11]. However, the use of technologies for its microencapsulation or nanoencapsulation has emerged to solve these drawbacks [13].

The objective of this study was to present key considerations for the design and development of geraniol-loaded nanocapsules for its delivery in the drinking water or feed of broiler chickens, and to evaluate its potential as an antimicrobial agent against *S*. Infantis using a standard in vitro microplate assay and a model that simulates the pH and feed conditions of the crop of broiler chickens. This approach aims to ensure efficacy in in vivo models, taking into account the principles of the “FDA Modernization Act 2.0 (29 December 2022)”, as well as the provisions of the European Parliament and the Organization for Economic Cooperation and Development (OECD) in 2021, which, although originally intended for human applications, could be adapted for use in animals [14,15].

## 2. Materials and Methods

### 2.1. Bacterial Strain and Culture Conditions

The CDC H3517 strain of *Salmonella* enterica subsp. enterica serovar Infantis (*S*. Infantis), previously described and selected for resistance to nalidixic acid (NA, 64 μg/mL, catalog no.N-4382, Sigma, St Louis, MO, USA), was used in the present study [16,17]. Briefly, 100 μL of *S*. Infantis from a frozen aliquot was placed in 10 mL of tryptic soy broth (TSB, Catalog No. 22092, Sigma, St Louis, MO, USA) and incubated at 37 °C overnight. Afterward, a subculture was prepared by inoculating 50 µL into 10 mL fresh TSB and incubated for 2 h to achieve log-phase growth (~6 log CFU/mL). This concentration of *S*. Infantis was verified by serial dilutions and plating on Xylose Lysine Tergitol-4 agar base (XLT-4, Catalog No. 223410, BD Difco™, Sparks, MD, USA) with NA for enumeration of real colony-forming units (CFU) in each assay.

### 2.2. Antimicrobial Activity of Geraniol Against S. Infantis by a Microplate Method

To establish the minimum inhibitory concentration (MIC) of geraniol (98%, 163333, Sigma-Aldrich, St Louis, MO, USA) on *S*. Infantis, a geraniol stock was first prepared at a concentration of 20,000 ppm using TSB with 2% Tween 80 (USP grade, Drogueria Cosmopolita, Naucalpan, Mexico City, Mexico) to promote the homogeneous dispersion of the essential oil. Subsequently, 100 µL of the suspensions with concentrations of 500, 1000, 2000, 4000, 6000 and 8000 ppm of geraniol in TSB with 2% Tween 80 were added in 96-well flat-bottom Bacti plates (microplate method, Catalog No. 269787; Nalgene Nunc International, Rochester, NY, USA), followed by 100 µL of a bacterial suspension with a concentration of 4.3 log CFU/mL, to reach a concentration of 4 log CFU/mL in each well. The geraniol concentrations evaluated were 250, 500, 1000, 2000, 3000 and 4000 ppm. TSB with 2% Tween 80 was used as control. The plates were incubated for two hours at 37 °C with orbital shaking (19 rpm; VWR, Houston, TX, USA). Finally, to evaluate the antimicrobial activity, 10-fold dilutions were made, cultured on XLT-4 selective medium with NA, and incubated at 37 °C for 18–24 h to enumerate total *S*. Infantis log CFU/mL. The reduction in S. infantis counts was determined by the difference between the counts (log CFU/mL) of the control group and the treatments. Each test was performed in triplicate.

### 2.3. Development and Selection of Geraniol Formulations Based on Polymeric Nanocapsules

The polymeric nanocapsules containing geraniol were prepared in two steps: (1) the formation of a nanoemulsion (core) by magnetic stirring (1000 rpm, IKA RO 10 Power, IKA-Werke GmbH & Co. KG, Staufen, Germany) at room temperature for one min between the oil phase consisting of geraniol, Miglyol^®^ 840 (Propylene Glycol Dicaprylate/Dicaprate, Dynamit Nobel, Witten, Germany) and an 80:20 surfactant mixture of Tween 80 and Span 80 (USP grade, Drogueria Cosmopolita, Naucalpan, Mexico City, Mexico), and the aqueous phase consisting of a 0.2 M phosphate buffer pH 6.0; and (2) deposition of Drugcoat E PO (Vikram Thermo, Gujarat, India) as a shell-forming polymer onto the core through electrostatic interactions. The nanocapsular system described in this study is based on a method previously reported by our research group [18], with the added advantage of eliminating the need for organic solvents.

The formulation selection was carried out through a 3^k^ factorial experimental design optimized to 15 formulations with 3 central points for validation (Appendix A) considering as factors Miglyol 840^®^ (0.5, 1, 1.5%, *v*/*v*), a 80:20 mixture of Tween 80:Span 80 (1.5, 2 and 2.5%, *v*/*v*) and Drugcoat E PO (5, 10 and 15 mg/mL). Geraniol was kept constant at 1% (*v*/*v*), and the total volume of the formulation was 5 mL. The response variables considered were particle size (nm), polydispersity index (PDI), zeta potential (mV) and encapsulation efficiency (EE).

### 2.4. Physicochemical Characterization

The geraniol-loaded nanocapsular systems of the experimental design were characterized in terms of particle size, PDI and zeta potential with the help of a Malvern Zetasizer Instrument (ZetaSizer Pro, Malvern Instruments, Worcestershire, UK).

#### 2.4.1. Particle Size and PDI

Particle size and polydispersity index were determined by dynamic light scattering (DLS) after dilution of the formulations in 0.2 M phosphate buffer pH 6.0 (1:40) to maintain the characteristics of the nanocapsular systems, and reduce viscosity and scattering effects. The analysis was performed in a disposable DTS1070 cell at 25 °C using a backscatter detection angle of 173° and considering refraction and absorption indices of 1.30 and 0.001, respectively. In each determination, 11 runs were performed to obtain a stable reading.

#### 2.4.2. Zeta Potential

Electrokinetic potential of the formulations was determined using laser Doppler velocimetry (LDV), which calculates mean electrophoretic mobility values. Briefly, the formulations were appropriately diluted to avoid charge shifts (1:40 in 0.2 M phosphate buffer pH 6.0). The determinations were also performed at 25 °C with an equilibration period of 120 s. Triplicates of each sample were measured and each measurement comprised 11 runs to obtain a stable reading. Results were analyzed using the ZS Xplorer software version 3.2.1.11 (Malvern Panalytical Ltd., Malvern, UK).

#### 2.4.3. Encapsulation Efficiency

The quantification of geraniol in the formulations was performed by UV-Vis spectrophotometry (UV-vis spectrophotometer, Varian, Cary 100 Tablet, Mulgrave, Australia). For each formulation, 0.5 mL of the formulations were placed in 2 mL Eppendorf tubes, followed by the addition of 0.5 mL of 2 M HCl and 1 mL of acetonitrile (ACN) for complete separation of the formulation components from geraniol. These mixtures were vortexed for 15 s and subsequently centrifuged at 7500 rpm for 10 min at 20 °C (Microfuge R20, Beckman Coulter Life Sciences, Palo Alto, CA, USA). Afterward, 0.015 mL of the supernatant was placed in a 15 mL Falcon centrifuge tube, followed by the addition of 2.5 mL of a 50:50 mixture of ACN and 0.2 M phosphate buffer (pH 6.0), previously adjusted to a pH 1.2 with 2 M HCl. Finally, the resulting solution was vortexed for 15 s to ensure complete homogenization. Geraniol was detected spectrophotometrically at 210 nm.

Encapsulation efficiency (EE) was determined using a two-step quantification approach. For this purpose, total geraniol was first quantified in each of the liquid formulations using the previously described methodology. For geraniol encapsulated in the polymeric nanocapsular systems, the liquid formulations were frozen at −80 °C (DR refrigerator) and freeze-dried (LabConco Freezone 4.5, Kansas City, MO, USA) for 12 h at −49 °C and a vacuum of 1.370 mBar. Freeze-dried formulations were rehydrated in the appropriate volume (considering the volume of freeze-dried formulation), and processed identically to total geraniol. The encapsulation percentage was determined using the following equation:$$\%EE=\frac{A}{B}\times 100$$
where A is the concentration of geraniol in the freeze-dried formulations and B the total concentration of geraniol in the liquid formulations.

### 2.5. Evaluation of the Antimicrobial Activity of the Formulations by Microplate Assays

The antimicrobial capacity of the two best formulations obtained (F3 and F9), considering average particle size (≤125 nm), PDI less than 0.1, and greater encapsulation efficiency, was evaluated in 96-well flat-bottom Bacti plates. The geraniol concentrations evaluated were 1000 and 3000 ppm, and the formulation corresponding to 3000 ppm but without geraniol was considered the blank. To achieve the above concentrations, dilutions of the formulations in TSB were made. The concentration of *S*. Infantis involved in the study was approximately 4 log CFU/mL. Once the formulations were in contact with *S*. Infantis, the plates were incubated for 2 h at 37 °C with orbital shaking (20 rpm; VWR, Houston, TX, USA). Subsequently, 10-fold dilutions were made, cultured on XLT-4 selective medium with NA and incubated at 37 °C for 12 h to enumerate total *S*. Infantis CFU. In this evaluation, the control was TSB with 2% Tween 80 and the blank was F3 without geraniol. Each test was performed in triplicate.

### 2.6. Evaluation of the Antimicrobial Activity of the Formulations Using a Model That Simulates the Crop of Broiler Chickens

An in vitro model was developed to simulate the pH and feeding conditions of the crop of broiler chickens, following previously published protocols [19], with slight modifications. For this purpose, 3 g of commercial feed (Nutricion Tecnica Animal SA. de CV., Queretaro, Mexico) containing 26% protein (12.64 MJ/kg metabolizable energy) and 10 mL of 0.1 M citrate buffer pH 5.0 were placed in 50 mL polypropylene centrifuge tubes. Subsequently, the corresponding volumes of the formulations to achieve geraniol concentrations of 300 and 3000 ppm were included in independent tubes, and a blank was also included for each concentration. Likewise, the corresponding volume of a bacterial suspension of *S*. Infantis was placed in the tubes to reach a concentration of 4 Log CFU/mL. Thereafter, the tubes were kept for 30 min in incubation at 40 °C under orbital shaking (19 rpm; VWR, Houston, TX, USA) and with an inclination of 30° to simulate the physiological conditions of the crop of broiler chickens and facilitate homogenization. After incubation, the contents of the tubes were homogenized by vortexing, 0.2 mL samples were taken and placed in 96-well flat-bottom Bacti plates for subsequent 10-fold dilutions, cultivation on XLT-4 selective medium with NA and incubation at 37 °C for 12 h to enumerate total *S*. Infantis CFU. For this evaluation, the control was TSB with 2% Tween 80, and the blanks consisted of F3 and F9 without geraniol. Each test was performed in triplicate.

### 2.7. Interaction Studies of the Formulation Components

#### 2.7.1. Fourier Transform Infrared Spectroscopy with Attenuated Total Reflectance (FTIR-ATR)

The interaction studies between the components that made up the nanocapsular systems were carried out using FTIR-ATR and only considering the proportions of each of these in F9 since it was the formulation that presented the smallest particle size and the one that contained the largest amount of the mixture of Tween 80:Span 80 and Miglyol^®^ 840, compared to F3. In this sense, FTIR spectra of geraniol, Miglyol^®^ 840, 80:20 mixture of Tween 80:Span 80, and Drugcoat E PO, as well as their mixtures such as Geraniol:Tween 80:Span 80, Miglyol^®^ 840:Tween 80:Span 80, Geraniol:Tween 80:Span 80:Drugcoat E PO, Miglyol^®^ 840:Tween 80:Span 80:Drugcoat E PO and the complete formulation were acquired using a FTIR NIR/MIR spectrophotometer (Perkin Elmer, Norwalk, CA, USA) equipped with a Miracle Diamond ATR unit, with a scanning range of 4000 to 450 cm^−1^, an average of 32 scans and a resolution of 4 cm^−1^. For each sample, a background spectrum was created and a baseline correction was performed for comparison and to establish possible interactions.

#### 2.7.2. Antimicrobial Activity in the Model That Simulates the Crop of Broiler Chickens

The antimicrobial capacity of mixtures of geraniol—Tween 80:Span 80, Miglyol^®^ 840—Tween 80:Span 80, geraniol—Miglyol^®^ 840—Tween 80:Span 80, geraniol—Tween 80:Span 80—Drugcoat E PO, Miglyol^®^ 840—Tween 80:Span 80—Drugcoat E PO and geraniol—Miglyol^®^ 840—Tween 80:Span 80—Drugcoat E PO (Formulation 9) was evaluated in the in vitro model simulating the crop of broiler chickens under the same considerations as for the complete formulations in the standard microplate assay, adjusting the geraniol concentrations to 3000 ppm. The control used in this evaluation was citrate buffer pH 5.0. Each test was performed in triplicate.

### 2.8. Statistical Analysis

Bacterial counts (*S*. Infantis Log CFU/mL) in the different assays were subjected to analysis of variance (ANOVA), followed by a Tukey post hoc test, respectively, to assess differences between groups (*p* < 0.05) using GraphPad Prism version 10.4.2 (GraphPad Software, San Diego, CA, USA). Data were expressed as mean ± standard error (SE). Before ANOVA, compliance with normality (Shapiro–Wilk test) and homogeneity of variances (Leven test) were assessed using Statgraphics Centurion XV (StatisticalGraphics Co., Rockville, MD, USA). The 3^k^ factorial experimental design and its evaluation were also performed in Statgraphics Centurion XV. All effects were considered significant at *p* < 0.05.

## 3. Results

### 3.1. Antimicrobial Activity of Geraniol Against S. Infantis by a Microplate Method

The results of the antimicrobial activity of different concentrations of geraniol against *S*. Infantis are shown in Figure 1. The results show significant differences starting from geraniol concentrations of 500 ppm compared to the control (TSB with 2% Tween 80). In fact, the most pronounced significant differences begin at the geraniol concentration of 3000 ppm, since it was able to reduce *S*. Infantis counts by more than half (3.35 Log CFU/mL) compared to the control. Notably, at a geraniol concentration of 4000 ppm, *S*. Infantis counts were completely reduced (6.22 Log CFU/mL), establishing this concentration as the MIC for subsequent trials.

### 3.2. Development and Selection of Geraniol Formulations Based on Polymeric Nanocapsules

Appendix A shows the experimental matrix of the 3^k^ factorial design considering the coded and uncoded levels of each factor, but the uncoded matrix was used for the design evaluation. Likewise, Appendix A shows the results of the response variables of each formulation (particle size, PDI, z-potential, and EE), highlighting that formulations 3, 10, and 13 were the midpoints of the design for validation.

The effects of each factor (%) evaluated in the experimental design considering the coefficients of the model of each variable studied are shown in Figure 2. The particle size of the formulations ranged from 73.6 to 273.0 nm. Factors such as Drugcoat E PO, Tween 80:Span 80 and the quadratic effect of Miglyol^®^ 840 showed a significant positive effect on particle size (*p* < 0.05), whereas Miglyol^®^ 840 and the Miglyol^®^ 840—Drugcoat E PO effect had a negative effect, that is, they tended to significantly reduce the particle size (*p* < 0.05) (Figure 2A). The model equation was as follows: Particle Size = 132.494−82.6786 (Miglyol^®^ 840) + 10.75 (Tween 80:Span 80) + 28.0625 (Drugcoat E PO) + 26.5411 (Miglyol^®^ 840)^2^ − 19.5 (Miglyol^®^ 840) (Drugcoat E PO). The adjusted determination coefficient (adjusted R^2^) of the model was 94.77%, which indicated that the model adequately explained the variation of the model and was therefore able to efficiently indicate the conditions for obtaining a given particle size.

In the case of PDI, its values were between 0.03 and 0.27 and the factors that significantly modified it upwards were Tween 80:Span 80 and the quadratic effect of Miglyol^®^ 840 (Figure 2B). Meanwhile, Miglyol^®^ 840 and the effect between Miglyol^®^ 840—Drugcoat E PO showed a negative effect on the PDI, that is, a significant reduction in PDI. The model presented an adjusted R^2^ of 78.67% and its equation was as follows: PDI = 0.0488036 − 0.0201036 (Miglyol^®^ 840) + 0.039825 (Tween 80:Span 80) + 0.0282625 (Drugcoat E PO) + 0.0509036 (Miglyol^®^ 840)^2^ − 0.029525 (Miglyol^®^ 840) (Tween 80:Span 80 − 0.056525 (Miglyol^®^ 840) (Drugcoat E PO).

The model that describes the variability of the zeta potential showed an adjusted R2 of 97.18%, presenting as upward factors the Drugcoat E PO, the quadratic effect of the Drugcoat E PO and the Miglyol^®^ 840—Drugcoat E PO interaction, which is expected since the Drugcoat E PO is the outermost part of the nanocapsule and, therefore, the one that should be relating to the zeta potential estimation (Figure 2C). The equation obtained from the model related to this variable was as follows: Zeta Potencial = 13.5 − 1.7625 (Miglyol^®^ 840 − 1.0375 (Tween 80:Span 80) + 1.2 (Drugcoat E PO) + 0.525 (Miglyol^®^ 840) (Tween 80:Span 80 + 0.575 (Drugcoat E PO)^2^.

Finally, for % EE only, the Drugcoat E PO was the factor that presented a significant effect increasing the encapsulation efficiency (Figure 2D). However, the adjusted R^2^ was 56.31%, which is relatively low for explaining the variability of the model: %EE = 31.3838 − 3.425 (Miglyol^®^ 840 + 1.70945 (Tween 80:Span 80) + 7.64055 (Drugcoat E PO + 3.45 (Miglyol^®^ 840) (Tween 80:Span 80) − 3.5 (Miglyol^®^ 840) (Drugcoat E PO); however, the central points were consistent in terms of the results obtained.

Considering the results of the design of experiments, two formulations were selected for presenting an appropriate particle size (less than 125 nm) and a PDI less than 0.20, formulation 3 (F3) and formulation 9 (F9), since F3 had a particle size of 122.0 nm, a low PDI (0.06), a zeta potential of 12.4 mV and %EE of 40.6, while formulation F9 had a smaller particle size (89.6 nm), but a higher PDI (0.12), a zeta potential of 11.0 mV and %EE of 49.2 (Appendix A). Although other formulations presented higher %EE such as F8, it was not selected for this experiment due to its larger particle size (273.0 nm) and higher PDI (0.27), which is related to a possible lower stability and lower size homogeneity of the nanocapsules, respectively.

### 3.3. Evaluation of the Antimicrobial Activity of the Formulations Obtained by Microplate Assays

The results of the antimicrobial activity of formulations F3 and F9 against *S*. Infantis evaluated in 96-well plates are shown in Figure 3. Both the control group and the blank formulations of F3 (F3-3000 B) and F9 (F9-3000 B) did not present significant differences, but the bacterial concentration increased by just over 2 Log CFU/mL after two hours of incubation. However, at a geraniol concentration of 1000 ppm in F3 (F3-1000) and F9 (F9-1000), the concentration of *S*. Infantis was significantly decreased by 1.70 and 1.59 Log CFU/mL, respectively, compared to the blank, but no differences were present between the formulations. Furthermore, at the concentration corresponding to 3000 ppm of geraniol in formulations F3-3000 and F9-3000, the bacterial concentration of *S*. Infantis was reduced by 2.92 and 2.87 Logs, respectively, compared to the control and the corresponding blank. These results indicate that the antimicrobial effect of the formulations is primarily attributable to geraniol.

### 3.4. Evaluation of the Antimicrobial Activity of Formulations in the Crop Assay

In addition to evaluating the antimicrobial activity of the formulations in microplates, their efficacy was also assessed using an in vitro model that simulates the conditions of the crop of broiler chickens in terms of pH and feed. The results of antimicrobial activity using this in vitro crop model are shown in Figure 4. After 30 min of exposure of formulations F3 and F9 with a complex matrix involving a pH of 5.0, feed and a concentration of *S*. Infantis of 4.31 Log CFU/mL, none of the geraniol concentrations (300 and 3000 ppm) in the selected formulations (F3 and F9) showed antimicrobial activity; in fact, there were no significant differences (*p* > 0.05). These results suggested an antagonistic effect between the formulation components and interaction between the components of the feed matrix that led to the nullification of the antimicrobial effect of geraniol. In this case, the concentration of *S*. Infantis was not modified since the incubation was for only 30 min and not 2 h as in the microplate assay.

### 3.5. Interaction Studies of the Formulation Components

#### 3.5.1. FTIR-ATR

The FTIR spectra of geraniol, Tween 80:Span 80, Miglyol^®^ 810N, Drugcoat E PO and their combinations in the proportions found in F9 are shown in Figure 5. In the FTIR spectrum corresponding to geraniol (Figure 5A), the stretching vibration in the band corresponding to the hydroxyl groups (O-H) is found at 3326 cm^−1^. Furthermore, the bands at 2959 cm^−1^ and 2909 cm^−1^ correspond to the stretching vibrations of -CH_2_. In the case of the band at 1670 cm^−1^, this corresponds to the angular deformation of C-O. At 1440 cm^−1^ and 1378 cm^−1^ are the bands associated with symmetric and non-symmetric vibrations of carboxylate group, respectively. Finally, the vibrational stretch of C=C is located at 996 cm^−1^. Meanwhile, in the surfactant mixture (Tween 80:Span 80, 70:30), the stretching vibrations in the bands at 3446 cm^−1^, 2926 cm^−1^ and 2859 cm^−1^, 1736 cm^−1^, and 1440 cm^−1^ and 1378 cm^−1^ are associated with O-H, -CH_2_, C-H, C=O (carbonyl) and C-O, respectively (Figure 5A). The FTIR spectrum of the combination of geraniol and Tween 80:Span 80 (28.6:71.4) shows a clear disappearance of the bands corresponding to the O-H, C-O, symmetric and non-symmetric vibrations of carboxylate and C=O of geraniol (Figure 5A), indicating possible hydrogen bonding or micellar encapsulation effects, which could hinder bioavailability. In fact, the FTIR spectrum of the combination closely resembles that of Tween 80:Span 80, suggesting a strong interaction between the two components.

On the other hand, the FTIR spectrum of Miglyol^®^ 810N (Figure 5B) displays five transmittance bands corresponding to the stretching vibrations of C-H (2936 cm^−1^, 2930 cm^−1^ and 2846 cm^−1^), C=O (1737 cm^−1^), and C-O-C (1135 cm^−1^). Interestingly, in the combination of Miglyol^®^ 810N and Tween 80:Span 80 (37.5:62.5), the stretching vibrations at 2936 cm^−1^ and 1135 cm^−1^ corresponding to Miglyol^®^ 810N are reduced in intensity and slightly shifted, indicating a mild interaction between these components (Figure 5B).

The FTIR spectrum of the mixture of geraniol, Miglyol^®^ 810N and Tween 80:Span 80 (20:30:50, Figure 5C) exhibits a pattern similar to those of the mixtures of geraniol and Tween 80:Span 80, and Miglyol^®^ 810N and Tween 80:Span 80, further confirming interactions among the components. Finally, in Figure 5D–F, it is clearly demonstrated that the Drugcoat E PO polymer is being deposited on the surface of the oily cores composed of geraniol and Tween 80:Span 80, Miglyol^®^ 810N and Tween 80:Span 80, and geraniol, Miglyol^®^ 810N and Tween 80:Span 80, confirming the successful formation of the nanocapsular system.

#### 3.5.2. In Vitro Crop Model to Evaluate the Antimicrobial Activity

Once the interactions between the components of F9 were identified by FTIR-ATR, the antimicrobial activity of the mixtures of geraniol—Tween 80:Span 80, Miglyol^®^ 840—Tween 80:Span 80, geraniol—Miglyol^®^ 840—Tween 80:Span 80, geraniol—Tween 80:Span 80—Drugcoat E PO, Miglyol^®^ 840—Tween 80:Span 80—Drugcoat E PO and geraniol—Miglyol^®^ 840—Tween 80:Span 80—Drugcoat E PO (Formulation 9) against *S*. Infantis was determined through an in vitro study simulating the crop of broiler chickens (Figure 6). The results demonstrate that there was a reduction in the antimicrobial activity of geraniol when it was present with Tween 80:Span 80, but it managed to significantly reduce the concentration of *S*. Infantis by 1.10 Log CFU/mL compared to the control. However, no other combination, including those containing geraniol, showed an antimicrobial effect, so a nullification of antimicrobial effects can be established due to the interaction between the formulation components and even the feed components, since in the microplate tests, F9 showed a better reduction in the concentration of *S*. Infantis.

## 4. Discussion

*S.* Infantis is an emerging zoonotic serovar of *Salmonella* that can be transmitted to humans through food, and is therefore considered a significant public health concern due to its recent rise in prevalence and dissemination in countries such as the USA, EU, and Latin America [2]. Although this bacterium has been isolated from pigs and cattle, it is most frequently found in broiler chickens, being the third most important *Salmonella* serotype [3,20], and, since 2014, in the EU, it is the main serotype isolated in poultry [1]. This bacterium has been reported to be a worldwide problem as it is associated with high levels of resistance to antibiotics, including third-generation cephalosporins and quinolones [2].

In this context, research into viable alternatives to antibiotics to combat bacterial resistance represents a real challenge. Among the alternatives being investigated is the group of essential oils since they have demonstrated excellent antimicrobial properties against both Gram-positive and Gram-negative bacteria [21]. However, some of the drawbacks that limit the use of essential oils are their poor stability due to their high volatility, rapid degradation when exposed to light and oxygen, low aqueous solubility, and intense aroma and flavor, the latter conditioning their inclusion into the feed or drinking water to treat or control bacterial diseases in animals [22]. Therefore, the antimicrobial activity of geraniol, both alone and in a nanocapsule-based formulation, was evaluated against *S*. Infantis using two in vitro models: one employing a standard microplate assay and the other simulating crop conditions in broiler chickens.

The results of this study demonstrated that at 500 ppm of geraniol, the concentration of *S*. Infantis was significantly reduced (0.55 Log CFU/mL) compared to the control (6.22 Log CFU/mL), and, at 4000 ppm, the growth of *S*. Infantis was completely inhibited (Figure 1). These results are consistent with those described in another study where significant inhibitions in the growth of *S*. Typhimurium were reported at 1000 ppm (0.1%) and complete inhibition at 5000 ppm (0.5%) using a broth microdilution method, with an initial *S*. Typhimurium concentration of 6.18 Log CFU/mL [11]. It has been described that the mechanism by which geraniol presents antimicrobial activity is related to its lipophilicity and/or hydrophobicity, but especially by the presence of the hydroxyl group found in its chemical structure, which causes membrane disruption, inhibition of oxygen uptake and alteration at the level of oxidative phosphorylation [23].

Based on these results, a nanocapsular system containing geraniol was developed and optimized through a design of experiments to enhance its stability and reduce its strong aroma and flavor. This was achieved by incorporating a flavor-masking and protective polymer (Drugcoat E PO, methacrylate co-polymer) for administration in the drinking water of broiler chickens. F3 and F9 exhibited the best characteristics in terms of size, PDI, zeta potential, and encapsulation efficiency (Appendix A). The encapsulation efficiency is particularly important, as the freeze-drying process enabled the production of a solid formulation, making it suitable for inclusion in the feed of broiler chickens. These formulations were evaluated in a microplate assay (Figure 3) and the results were similar to those of geraniol alone (Figure 1). This suggests that the antimicrobial effect was attributed to geraniol, rather than the other components of the formulation. At the pH of TSB (approximately 7.1), Drugcoat E PO remains insoluble but facilitates the permeation of geraniol [24].

In this context, the formulations were also evaluated in an in vitro model that simulates the pH and feed storage conditions of the crop of broiler chickens (Figure 4), as previously described by our research group [19]. However, the results were not as expected. Instead of reducing *S*. Infantis counts, the antimicrobial effect of geraniol was nullified, even at concentrations as high as 3000 ppm, suggesting a strong interaction between the formulation components and the feed.

Therefore, to corroborate these findings, interaction studies between the components of the formulation were conducted using FTIR analysis (Figure 5). The results suggested a strong correlation between geraniol, the Tween 80:Span 80 mixture and Miglyol^®^ 810N. In fact, in geraniol, one of the most important stretching vibrations corresponding to the hydroxyl groups (O-H, 3326 cm^−1^) [25,26,27] disappeared in the mixture of geraniol with Tween 80:Span 80 [28] (Figure 5A). This behavior is related to the decrease in antimicrobial activity in the in vitro model that simulates the crop (Figure 6), because although the reduction in S. Infantis was significant with respect to the control, it was only reduced by 1.10 Log CFU/mL. These results differ slightly from the microplate assay where *S*. Infantis counts were reduced by just less than 3 Log CFU/mL. Although both surfactants are non-ionic and widely used in the stabilization of emulsions/nanoemulsions, it has been reported that hydrophobic interactions can take place between these surfactants and essential oils, which limits the amount of essential oil available and then reduces its antimicrobial and antioxidant effects, but this effect is more noticeably observed as the proportion of surfactant is higher with respect to essential oil in a formulation [29,30]. Therefore, in the early stages of development of nanocapsular systems based on essential oils stabilized with surfactants, it is important to determine the FTIR behavior to evaluate interactions and thus select the most suitable one. Although slight interactions were observed between Miglyol^®^ 810N and the Tween 80:Span 80 mixture, and interactions of some fatty acids with essential oils have been reported [31,32,33], when the mixture of geraniol, Miglyol^®^ 810N and Tween 80:Span 80 was evaluated, no different behaviors were observed in the FTIR spectra of the two-component mixtures (geraniol—Tween 80:Span 80 and Miglyol^®^ 810N—Tween 80:Span 80). Therefore, these findings suggest that components within the complex feed matrix interacted with the formulation, thereby completely nullifying the antimicrobial activity of geraniol, as shown in Figure 6.

The present nanocapsular systems were obtained by an organic solvent-free method based on emulsification/deposition [34,35]. For this, a nanoemulsion was first obtained, which was stabilized with an 80:20 combination of Tween 80:Span 80, and subsequently, through electrostatic interactions, the Drugcoat E PO polymer was deposited onto the emulsion cores. The deposition of Drugcoat E PO was performed because at the pH of the formulation (pH 6), its alkyl amino group was protonated [24], which favored its interaction with the negatively charged nucleus due mainly to the carboxyl groups of Miglyol^®^ 810N. The formation of the nanocapsular systems was corroborated during the analysis of interactions in the FTIR spectrum, since in Figure 5D–F, it is clear that the behavior of the formulation is the same as that of Drugcoat E PO [36]. Additionally, the characteristic odor of geraniol was significantly reduced, supporting the successful encapsulation and masking effect.

## 5. Conclusions

The development of nanotechnological platforms as alternatives to antibiotics represents a promising strategy to address the challenges of bacterial resistance and public health concerns associated with pathogens such as *S*. Infantis. However, the effectiveness of these nanosystems is not solely determined by their favorable physicochemical properties and stability. Without proper preformulation studies, even well-characterized nanosystems may fail under real-world poultry farming conditions. Therefore, it is essential to conduct thorough preformulation evaluations to guide the appropriate selection of formulation components. Additionally, the use of in vitro models that closely simulate the physiological conditions of broiler chickens is crucial for reliably assessing the antimicrobial efficacy of such formulations since feed is a complex matrix that can also impact their effect. Further research is underway to explore alternative surfactants, aiming to enhance the performance of the geraniol-based nanocapsular system against *S*. Infantis.

This research sets an important precedent for the development of robust nanocapsular systems containing essential oils, with an emphasis on preformulation studies and in vitro models that simulate in vivo conditions and ensure their efficacy as antimicrobial agents. This is useful since many products are developed that do not exhibit the expected effect because they are evaluated in in vivo models or in the target species, thinking at the veterinary level, specifically in production species such as poultry, but without considering that they can also be applied in humans.

## Figures and Tables

**Figure 1 pharmaceutics-17-00840-f001:**
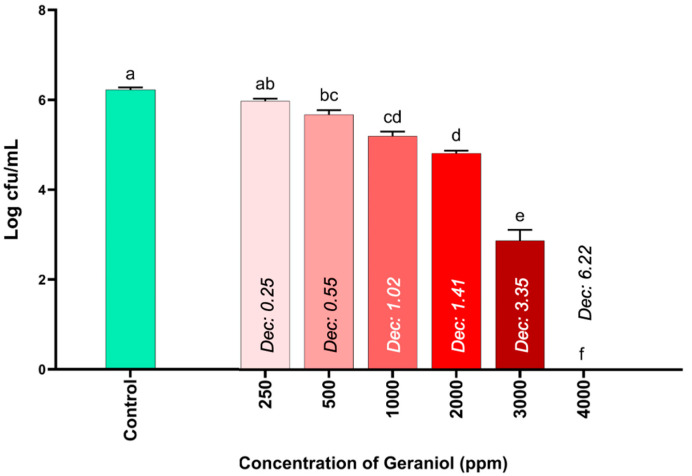
Evaluation of the antimicrobial activity of geraniol against *S*. Infantis using an in vitro microplate model. The results are expressed as the mean ± SE. ^a–f^ Bars with different letters are considered significant (*p* < 0.05). Each experimental group was evaluated in triplicate (*n* = 3).

**Figure 2 pharmaceutics-17-00840-f002:**
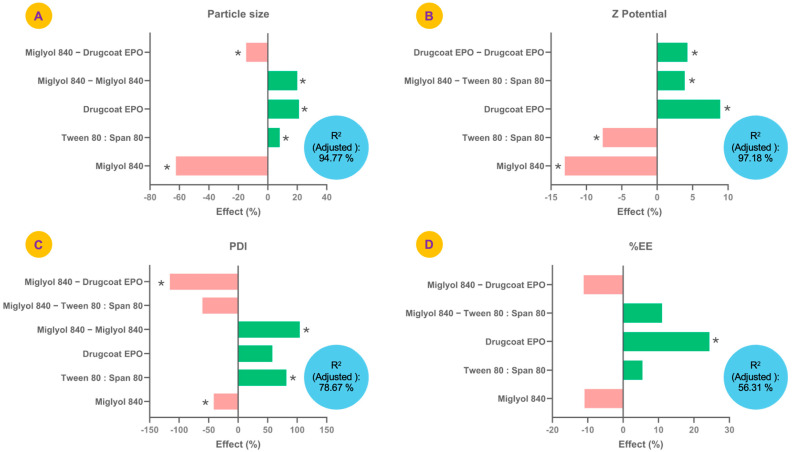
Effects of the factors studied in the design of experiments on the response variables, (**A**) particle size (nm); (**B**) Z-potential (mV); (**C**) polydispersity index (PDI); and (**D**) geraniol encapsulation efficiency (%), used to optimize the formulation of geraniol polymeric nanocapsules. * indicates significant effects (*p* < 0.05).

**Figure 3 pharmaceutics-17-00840-f003:**
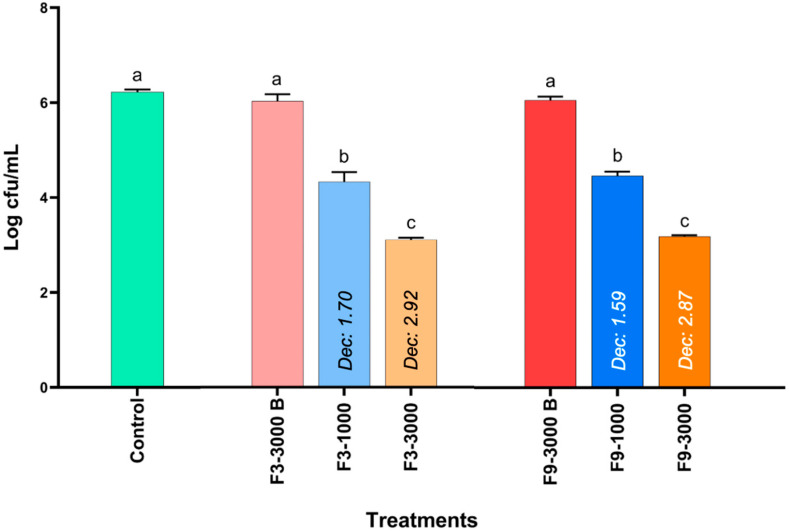
Determination of the antimicrobial activity of two geraniol formulations based on polymeric nanocapsular systems against *S*. Infantis through an in vitro microplate assay considering 1000 and 3000 ppm of geraniol. The results are expressed as the mean ± SE. ^a–c^ Bars with different letters are considered significant (*p* < 0.05). Each experimental group was evaluated in triplicate (*n* = 3). F3) Geraniol–Miglyol® 810N–Twen 80:Span 80 (1%–1%–2%) and F9) Geraniol–Miglyol® 810N–Twen 80:Span 80 (1%–1.5%–2.5%).

**Figure 4 pharmaceutics-17-00840-f004:**
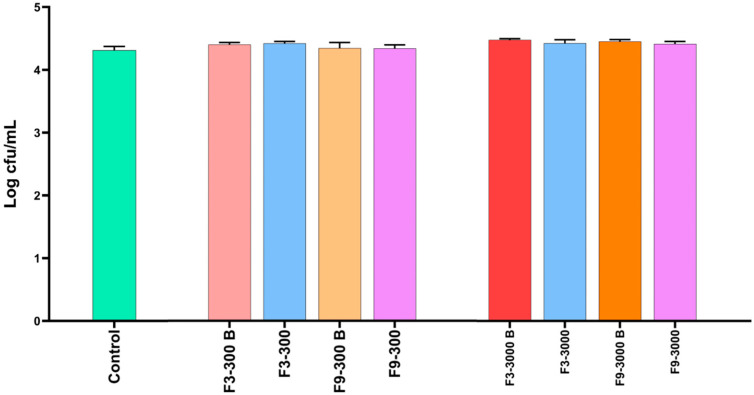
Antimicrobial activity of two geraniol formulations based on polymeric nanocapsules against *S*. Infantis in an in vitro model simulating crop conditions in terms of pH and presence of feed. The results are expressed as the mean ± SE. Each experimental group was evaluated in triplicate (*n* = 3).

**Figure 5 pharmaceutics-17-00840-f005:**
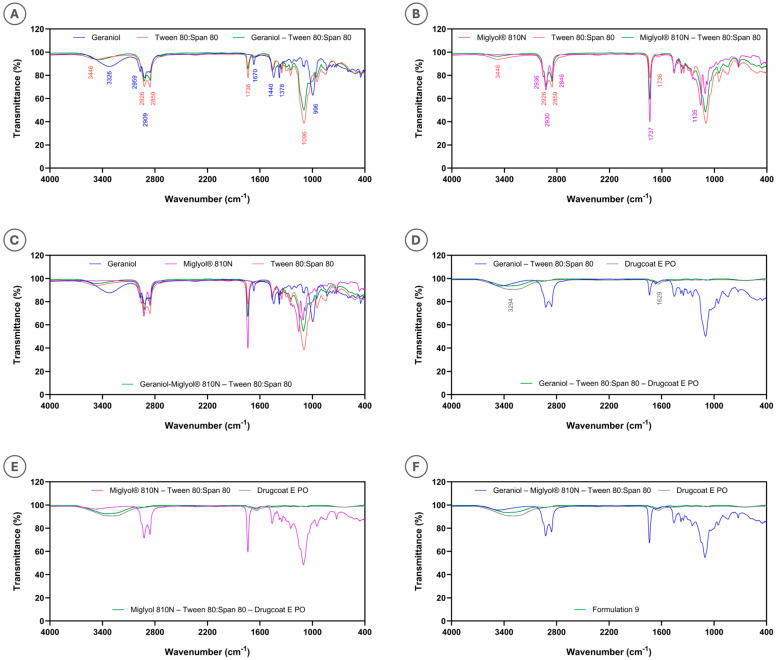
FTIR-ATR spectra of (**A**) geraniol, Tween 80:Span 80 and their combination; (**B**) Miglyol^®^ 810N, Tween 80:Span 80 and their combination; (**C**) geraniol, Miglyol^®^ 810N, Tween 80:Span 80 and their combination; (**D**) geraniol-Tween 80:Span 80 and Drugcoat E PO; (**E**) Miglyol^®^ 810N-Tween 80:Span 80 and Drugcoat E PO; and (**F**) full formulation.

**Figure 6 pharmaceutics-17-00840-f006:**
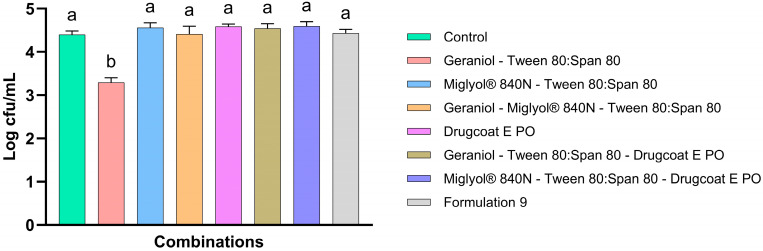
Antimicrobial activity of binary and ternary combinations of components in geraniol-based nanocapsular formulations. The results are expressed as the mean ± SE. ^a,b^ Bars with different letters are considered significant (*p* < 0.05). Each experimental group was evaluated in triplicate (*n* = 3).

## Data Availability

The databases used and analyzed during the current study are available from the corresponding author.

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
