# Peer review of "Design of Geraniol-Loaded Nanocapsules for Use Against Salmonella Infantis: Evaluation in an In Vitro Poultry Model"

_pharmaceutics, 2025, doi:10.3390/pharmaceutics17070840_

Round 1
Reviewer 1 Report
Comments and Suggestions for Authors
This manuscript reviews the possible applications of geraniol-containing nanocapsules in antimicrobial activities. Due to the problem of antibiotic resistance, it is interesting to explore future molecules as anti-infective agents.
The manuscript is well structured and contains the most important information on the antimicrobial properties of this product.There are some aspects that should be improved:
- the authors in the introduction refer to U.S. law - specifically the Food and Drug Administration (FDA) - so it would be useful to also relate the news to the other major EU market and the EU law applicable there.
- there is a lot of information in the abstract, but some data are missing, e.g. what is the main mechanism of antimicrobial action?
- in the methods section when describing each method, I would add a sentence at the end of the description:
positive control is..., negative control is ....i how many times the experiment was repeated.
4) I would suggest adding a short statement identifying the unique contribution of this review, which would help emphasize its importance and originality.
5) It would be helpful to highlight how the manuscript contributes to the field.
Author Response
We thank you very much for the time you have spent on reviewing our manuscript. We have given full consideration to your comments and the manuscript that has been carefully revised and modified accordingly. Please refer to the point-by-point reply to your comments
- Comment 1
This manuscript reviews the possible applications of geraniol-containing nanocapsules in antimicrobial activities. Due to the problem of antibiotic resistance, it is interesting to explore future molecules as anti-infective agents.
The manuscript is well structured and contains the most important information on the antimicrobial properties of this product.There are some aspects that should be improved:
The authors in the introduction refer to U.S. law - specifically the Food and Drug Administration (FDA) - so it would be useful to also relate the news to the other major EU market and the EU law applicable there.
Response
Information from the EU on the reduction in the use of experimental animals in 2021 was included in L.99-101. Thank you.
- Comment 2
There is a lot of information in the abstract, but some data are missing, e.g. what is the main mechanism of antimicrobial action?
Response
In L.40-41, the main mechanism by which geraniol carries out its antimicrobial effect was included. Furthermore, in L.490-491 the mechanism of action of geraniol was complemented. Thank you.
- Comment 3
In the methods section when describing each method, I would add a sentence at the end of the description:
positive control is..., negative control is ....i how many times the experiment was repeated.
Response
In L.131-132, L.216-217, L.236-237 and L.262-263 the experimental groups and the number of trials were included at the end of the methodologies. Thank you.
- Comment 4
I would suggest adding a short statement identifying the unique contribution of this review, which would help emphasize its importance and originality.
Response
In L.567-573 a paragraph was included highlighting the contribution of this research. Thank you.
- Comment 5
It would be helpful to highlight how the manuscript contributes to the field.
Response
In L.567-573, the contribution of this research to the veterinary field was highlighted. Thank you.

Reviewer 2 Report
Comments and Suggestions for Authors
Dear Authors,
in your manuscript you presented the design of nanocapsules loaded with geraniol and its effect on S. Infantis by testing it with two in vitro model.
general comments:
- please change the title, it is too long
-It is difficult to give specific comments as there are no line numbers
-please check writing of Drugcoat E PO in the whole text
Abstract -please rephrase: "the objective of the present study was to present.."; "their potential an antimicrobial agent"; "an standard"; "and not only consider"; "is frequently considered"
Introduction - please rephrase "can be important through water"; so its prevalence has increased"; "new viable alternatives to their use"; because after the systematic review it was the one" - this sentence is also not clear enough; "their potential an antimicrobial agent"; "an standard"; "feed conditions of the crop";
Materials and Methods - please change or rephrase: "Xylose Lysine Tergitol-4 what -agar?; cfu; how did you apply S. Infantis on the Bacti plates (chapter 2.2. is not clear enough, please rephrase); Chapter 2.4. starts with almost the same sentence as the 2.3. ends with; chapter 2.4.2. please mention the manufacturer; chapter 2.6. please change this title and in the whole text uniform the writing of the crop model (broiler chicken crop or crop of broiler chickens); not clear -300 and 3000 ppm were included in the same tubes? Chapter 2.7.1. please change the beginning, not clear enough?; chapter 2.7.2.- difference between this one and 2.6.?
Results - chapter 3.2. please change first sentences as these are more appropriate for Materials and Methods; Drugoat? please shorten the title of 3.4.; "evaluated presented antimicrobial activity?"; twice and and; 3.5.2. can be established?
Discussion - please rephrase: "their derivates"; please rephrase first two sentence in the second paragraph "in this context..."; please check writing of S. typhimurium; twice were were; please shorten the sentence "in fact, in geraniol..."; please change the last paragraph (Finally, to obtain...").
Comments on the Quality of English Languageminor editing needed
Author Response
We thank you very much for the time you have spent on reviewing our manuscript. We have given full consideration to your comments and the manuscript that has been carefully revised and modified accordingly. Please refer to the point-by-point reply to your comments
In your manuscript you presented the design of nanocapsules loaded with geraniol and its effect on S. Infantis by testing it with two in vitro model.
general comments:
- Comment 1
- Please change the title, it is too long
Response
The title of the manuscript was changed to "Design of geraniol-loaded nanocapsules for use against Salmonella Infantis: evaluation in an in vitro poultry model". Thank you.
- Comment 2
-It is difficult to give specific comments as there are no line numbers
Response
We apologize for the inconvenience, something probably happened while uploading our manuscript when submitting it. Thank you.
- Comment 3
-please check writing of Drugcoat E PO in the whole text
Response
The writing of the Drugcoat E PO was reviewed throughout the manuscript. Thank you.
- Comment 4
Abstract -please rephrase: "the objective of the present study was to present.."; "their potential an antimicrobial agent"; "an standard"; "and not only consider"; "is frequently considered"
Response
In L.33-38 is the reworded objective. Thank you.
- Comment 5
Introduction - please rephrase "can be important through water"; so its prevalence has increased"; "new viable alternatives to their use"; because after the systematic review it was the one" - this sentence is also not clear enough; "their potential an antimicrobial agent"; "an standard"; "feed conditions of the crop";
Response
In L.93-97, adjustments were made to the wording taking into account the suggestions. Thank you.
- Comment 6
Materials and Methods
- please change or rephrase: "Xylose Lysine Tergitol-4 what -agar?
Response
In L113, the corresponding change was made to Xylose Lysine Tergitol-4 agar base (XLT-4, Catalog No. 223410, BD DifcoTM). Thank you.
- cfu
Respose
In L.114 the meaning of CFU (colony-forming units) was included. Thank you.
- how did you apply S. Infantis on the Bacti plates (chapter 2.2. is not clear enough, please rephrase)
Respose
The methodology is described in more detail in L.121-126. Thank you.
- Chapter 2.4. starts with almost the same sentence as the 2.3. ends with
Response
We understand the concern, but in chapter 2.3 reference is made to the response variables evaluated in the design of experiments and in chapter 2.4 characterization is mentioned, which is important to evaluate the design. Thank you.
- chapter 2.4.2. please mention the manufacturer
Response
In L.178, the manufacturer was included. Thank you.
- chapter 2.6. please change this title and in the whole text uniform the writing of the crop model (broiler chicken crop or crop of broiler chickens)
Response
In L.222, 232 and 233, the way in which "the crop of broiler chickens" is presented was standardized. Thank you.
- not clear -300 and 3000 ppm were included in the same tubes?
Response
Apologies for the confusion, the corresponding changes were included in L.228-229. Thank you.
- Chapter 2.7.1. please change the beginning, not clear enough?;
Response
The wording was modified as suggested. Thank you.
- chapter 2.7.2.- difference between this one and 2.6.?
Response
The model is exactly the same, but the objective was different since in chapter 2.7.2 the aim was to demonstrate which of the interactions between components of the formulation presented a nullification of the antimicrobial effect of geraniol. Thank you.
- Comment 6
Results - chapter 3.2. please change first sentences as these are more appropriate for Materials and Methods.
Response
The first sentence was modified according to the suggestion. Thank you.
- Drugoat?
Response
Thank you very much for the comment. It has already been corrected in L.250, 261, 298, 314, 321, 333, 452, 502, 510 and 553. Thank you.
- please shorten the title of 3.4.;
Response
The title was changed to "3.4. Evaluation of the antimicrobial activity of formulations in the crop assay. Thank you.
- "evaluated presented antimicrobial activity?";
Response
The wording of the sentence was corrected. Thank you.
- twice and and; 3.5.2. can be established?
Response
Thank you very much for the observation, it was corrected in L.431 and 444.
- Comment 7
Discussion
- please rephrase: "their derivates";
Response
The sentence was corrected according to the suggestion. Thank you.
- please rephrase first two sentence in the second paragraph "in this context...";
Response
The sentence was rewritten according to the suggestion. Thank you.
- please check writing of S. typhimurium;
Response
It was corrected throughout the manuscript. Thank you.
- twice were were;
Response
Thank you very much for the observation (L.518). Thank you.
- please shorten the sentence "in fact, in geraniol...";
Response
The sentence was reduced and given meaning. Thank you.
- please change the last paragraph (Finally, to obtain...").
Response
The sentence was rewritten. Thank you.

Round 2
Reviewer 2 Report
Comments and Suggestions for Authors
Dear Authors,
thank you for taking into consideration the comments and suggestions given by the reviewer.
There is still a need for minor changes in L 131, 215-216, 235 and 263-264 - please write those sentences in full, as they are currently not fully appropriate.
Author Response
Dear reviewer,
We appreciate you taking the time to improve the quality of our manuscript.
The suggested corrections have been made and you can see them highlighted in blue.
The sentece “Control: TSB with 2% Tween 80” in L.132 was deleted since in L.126-127. was described what the control consisted of.
The sentence in L.216-218 was modified according to the suggestions.
The sentence in L.237-238 was modified according to the suggestions.
The sentence in L.265-266 was modified according to the suggestions.
